# Research on SUnet Winter Wheat Identification Method Based on GF-2

Ke Zhou [1,2], Zhengyan Zhang [1,2], Le Liu [1,2], Ru Miao [1,2,*], Yang Yang [1,2], Tongcan Ren [1,2] and Ming Yue [1,2]

[1] School of Computer and Information Engineering, Henan University, Kaifeng 475004, China
[2] Henan Province Engineering Research Center of Spatial Information Processing, Henan University, Kaifeng 475004, China
* Correspondence: mr1015@henu.edu.cn

**Abstract:** Introduction: Winter wheat plays a crucial role in ensuring food security and sustainable agriculture. Accurate identification and recognition of winter wheat in remote sensing images are essential for monitoring crop growth and yield estimation. In recent years, attention-based convolutional neural networks have shown promising results in various image recognition tasks. Therefore, this study aims to explore the application of attention-based convolutional neural networks for winter wheat identification on GF-2 high-resolution images and propose improvements to enhance recognition accuracy. Method: This study built a multi-band winter wheat sample dataset based on GF-2 images. In order to highlight the characteristics of winter wheat, this study added two bands, NDVI and NDVI$_{increase}$, to the dataset and proposed a SUNet network model. In this study, the batch normalization layer was added to the basic structure of the UNet convolutional network to speed up network convergence and improve accuracy. In the jump phase, shuffle attention was added to the shallow features extracted from the coding structure for feature optimization and spliced with the deep features extracted by upsampling. The SUNet made the network pay more attention to the important features to improve winter wheat recognition accuracy. In order to overcome the sample imbalance problem, this study used the focus loss function instead of the traditional cross-entropy loss function. Result: The experimental data show that its mean intersection over union, overall classification accuracy, recall, F1 score and kappa coefficient are 0.9514, 0.9781, 0.9707, 0.9663 and 0.9501, respectively. The results of these evaluation indicators are better than those of other comparison methods. Compared with the UNet, the evaluation indicators have increased by 0.0253, 0.0118, 0.021, 0.0185, and 0.0272, respectively. Conclusion: The SUNet network can effectively improve winter wheat recognition accuracy in multi-band GF-2 images. Furthermore, with the support of a cloud platform, it can provide data guarantee and computing support for winter wheat information extraction.

**Keywords:** winter wheat identification; SUNet; remote sensing image

## 1. Introduction

As one of the three major grain crops in China, wheat occupies an important position in the daily life of the people and the national economy [1]. Accurately identifying winter wheat from high-resolution images requires remote sensing technology and convolutional neural networks [2,3]. This is paramount to winter wheat spatial distribution accuracy and ensuring grain yield [4].

Convolutional neural networks (CNN) can automatically extract abstract features and reduce the number of parameters in the neural network during the deep propagation process [5]. It is widely used in image recognition [6–10], object detection [11–13], semantic segmentation [14–18], remote sensing scene classification [19–21], image denoising [22–24] and other fields. The extensive application of convolutional neural networks in computer vision signifies that their techniques and methods can be effectively employed in remote

sensing [25,26] image automatic interpretation. The use of image semantic segmentation has become an important technology in order to extract the data regarding crops from high-resolution images. For example, Zhang et al. [27] constructed a hybrid structure convolutional neural network (HSCNN) with two different depth components based on CNN which was used to extract winter wheat planting information in the study area. Compared with SegNet [28] and DeepLab [29], their method improves accuracy by 0.147 and 0.059, respectively, and achieves good results. Chen [30] constructed a multi-scale feature convolution neural network (MSFCNN) model to extract winter wheat planting information in the study area. Furthermore, Wang et al. [31] proposed a method to improve the accuracy of image edge pixel extraction, thereby improving the classification accuracy of winter wheat. This approach used a partially connected conditional random field model (PCCRF) to refine RefineNet [32] classification results. The results showed that this method effectively extracted winter wheat spatial distribution information. In addition, Teimouri et al. [33] developed and implemented a lightweight network structure. This network structure is based on deep learning and combines a fully convolutional network and a convolutional long short-term memory network (ConvLSTM). The authors applied this network structure to the spatial information extraction of 14 crops in their study area. The average pixel-based accuracy and Intersection over Union obtained from the proposed network were 86% and 0.64, respectively, and achieved good results. In remote sensing image semantic segmentation, the encoder–decoder structure similar to that of the UNet networks to stitch low-level and high-level features achieved better results.

However, the current research primarily relies on Sentinel and Landsat data, with limited utilization of Chinese GF data. This is mainly due to the relative difficulty in accessing Chinese GF data. Chinese GF data is typically controlled and managed by the government or relevant organizations, requiring specific application procedures or appropriate permissions to access such data. The spatial resolution of GF-2 data in China can reach the sub-meter level. The GF-2 PMS data can provide optical images at a 0.8 m resolution, which greatly exceeds Sentinel (10 m) and Landsat (30 m) capabilities, thus providing information on winter wheat, e.g., texture and field shape. However, there are few public datasets of GF-2 winter wheat for training high-resolution images, and traditional machine learning methods still need to extract features artificially. As a result, only shallow features can be learned when training on high-resolution images, which affects the final target recognition accuracy. Furthermore, due to the different planting area sizes of winter wheat, there are different patch sizes in remote sensing images, which causes unbalanced samples in the dataset. Determining ways to construct a high-resolution remote sensing image dataset of winter wheat and use a convolutional neural network to accurately identify winter wheat information on high-resolution images are also problems to be solved.

This paper proposes a SUNet (Shuffle Attention UNet) network model combined with an attention mechanism. The primary purpose is to solve the problem of unbalanced sample datasets and high-resolution winter wheat extraction tasks. In order to make the neural network pay more attention to important features, a batch normalization layer is added to improve the target recognition accuracy after layer convolution. In the decoding stage, the underlying features of the encoding stage are optimized to strengthen the network model's area of interest. Then, this paper uses the focus loss function to calculate the loss value. This can alleviate the problem of sample imbalance. Thus, the classification accuracy of winter wheat can be improved. In addition, the data enhancement strategy can be used to expand the winter wheat dataset during training. This makes the network model more robust, and it also avoids overfitting.

In summary, the objectives of this study are as follows:

(1) Construct a high-resolution winter wheat public dataset based on the China GF-2 satellite. The dataset contains six bands of RGB, near-infrared, NDVI and $NDVI_{increase}$, and has rich image samples and labeling information.

(2)    Propose the SUNET network model, which introduces the Batch normalization layer and the Shuffle Attention mechanism. The results of the comparison test and the ablation experiment show that the generalization ability and classification accuracy of the SUNET model have been improved.

## 2. Methodology

### 2.1. Research Area and Data Source

#### 2.1.1. Research Area

This paper selected Kaifeng City, Henan Province as the research area. Kaifeng City is located in the east of Henan Province, between 34°11′43″–35°11′43″N and 113°51′51″–115°15′42″E, as shown in Figure 1. The east–west length is about 126 km, the north–south width is about 92 km, and the total area is 6266 square kilometers. Kaifeng City is located in the eastern part of the alluvial plain in the middle and lower reaches of the Yellow River, with a 69–78 m altitude and flat terrain. The climate is distinct in four seasons and belongs to a temperate monsoon climate. The general climate characteristics are high temperature and rainy weather in summer and cold and dry weather in winter. The annual average frost-free period is 221 days, the average yearly temperature is 14.52 °C, the annual average precipitation is 627.5 mm, and the precipitation is mainly concentrated in July and August in summer. Kaifeng City is rich in groundwater resources, and the natural ecological environment is suitable for the growth of various crops.

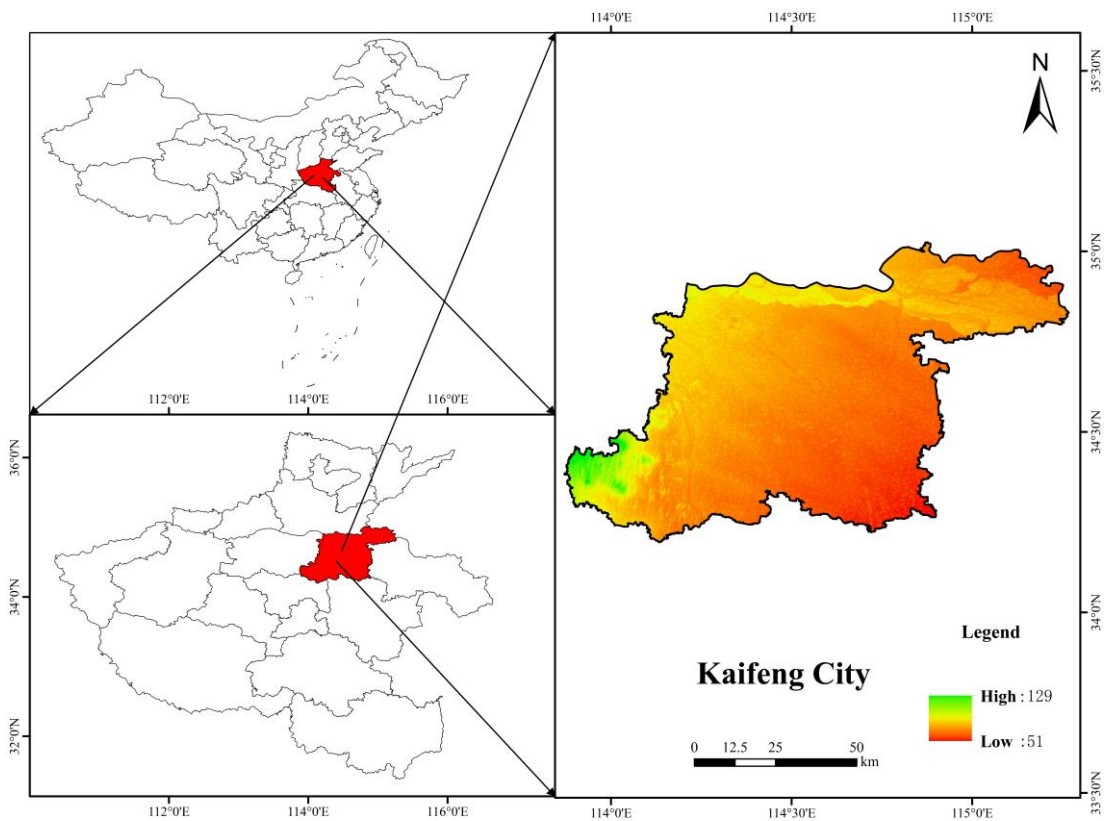

**Figure 1.** Study area location map.

#### 2.1.2. Data Source

GF-2 satellite is the first civil optical remote sensing satellite with a resolution better than 1 m independently developed by China. It was successfully launched on 19 August 2014. It was first imaged and downloaded on 21 August 2014. The detailed parameters of GF-2 are shown in Table 1. It has the characteristics of high-speed positioning, sub-meter spatial resolution and fast maneuvering side swing ability, which effectively improves the satellite's comprehensive observation ability. The image sample is shown in Figure 2.

At present, GF-2 satellite plays an important role in land and resource survey, urban fine management, mineral resource monitoring, crop growth monitoring and yield estimation, forest resource survey, desertification monitoring, geological disaster monitoring, flood monitoring and post-disaster reconstruction.

**Table 1.** GF-2-related parameter information.

| Sensor Type | Spatial Resolution | Band Number | Spectral Range | Width | Remark |
|---|---|---|---|---|---|
| Panchromatic | 1 m | 1 | 0.45~0.90 μm | | |
| | | 2 | 0.45~0.52 μm | | Blue |
| | | 3 | 0.52~0.59 μm | 45 km | Green |
| Multispectral | 4 m | 4 | 0.63~0.69 μm | | Red |
| | | 5 | 0.77~0.89 μm | | Near infrared |

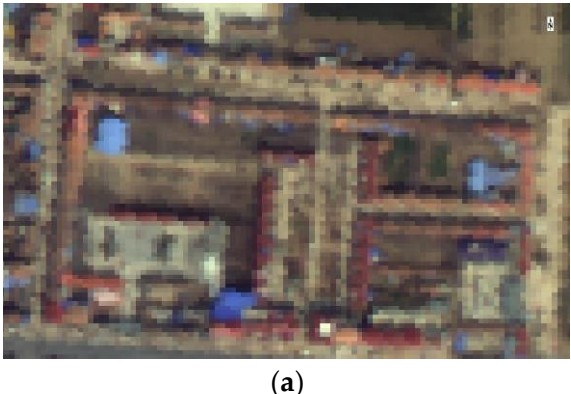 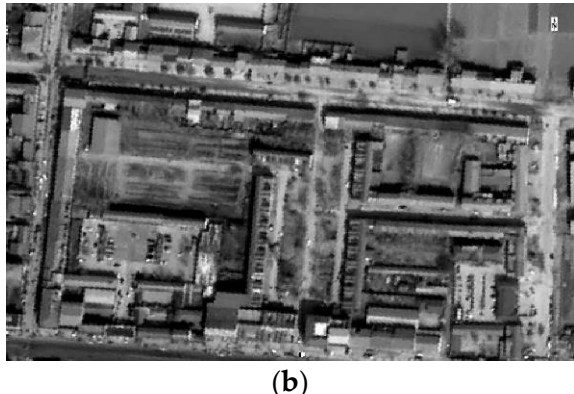

(**a**) (**b**)

**Figure 2.** GF-2 data sample. (**a**) Multispectral bands; (**b**) panchromatic band.

The experimental data used the GF-2 image data of the western region of Lankao County, Kaifeng City on 5 March 2020. During this period, winter wheat is in the rising and jointing stage, which is more obvious than other features [34]. Wheat seedlings are green in remote sensing images. In addition, GF-2 image data from 28 May in the same year and the same region were selected for comparison to improve the sample production accuracy of winter wheat. At this time, winter wheat was in the milk-filling stage, and wheat seedlings were brown in true color images. The two-image data selected in the experiment have less cloud content and higher clarity.

*2.2. Technical Process*

GF-2 image data were selected to extract winter wheat spatial distribution information in the study area in 2020 using a convolution neural network. At the same time, an improved UNet neural network was applied to improve winter wheat recognition accuracy. The process steps are as follows, as shown in Figure 3: Step 1—data preprocessing operations are performed on the GF-2 image, including radiometric calibration, atmospheric correction, orthorectification and image fusion. Then, the processed images are marked to make a winter wheat sample dataset; Step 2—the improved UNet convolutional neural network model is constructed; Step 3—the research on the accuracy of winter wheat recognition is realized using the improved convolutional neural network on the AI studio platform of Baidu; Step 4—the experimental results are analyzed by comparing the experiment and evaluation indicators.

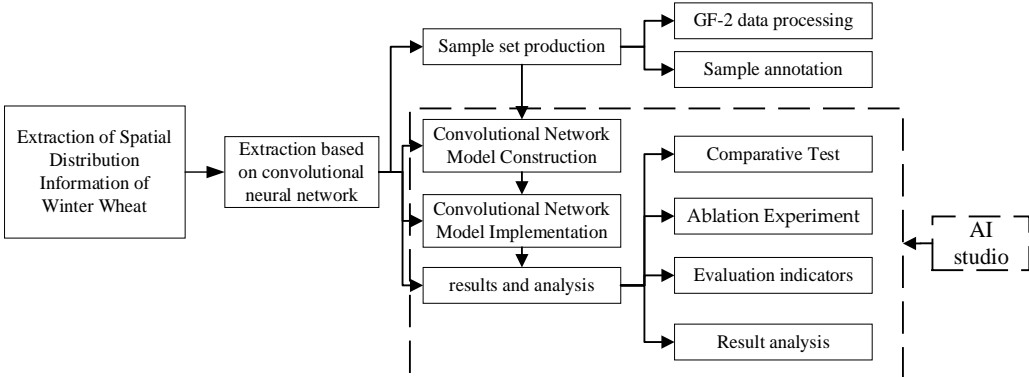

**Figure 3.** Technical flow chart.

*2.3. SUNet Network Construction*

　　UNet is a pixel-level image segmentation network that performs well on small sample datasets, and its function is to identify and label each pixel of the image [35]. The UNet network adopts a U-shaped symmetrical structure. The compression channel (encoder) and expansion channel (decoder) of convolution and pooling operation are on their left and right sides. The training set was extracted layer by layer through the encoding–decoding structure. The function of the encoder part of the network is to perform multi-scale feature recognition and extraction on the image matrix through four pooling operations. Feature fusion at different scales improves the classification accuracy of images. This paper extracted the high-resolution winter wheat data, improved the UNet model, and proposed the SUNet network model. A batch normalization layer (BN) was added after each convolution layer of the UNet network to improve the training speed and convergence speed and prevent overfitting. In this paper, the Shuffle Attention was used instead of traditional direct connection in the feature splicing stage of the UNet network decoding structure. Including the Shuffle Attention mechanism can strengthen the region of interest of the model and achieve selective and full feature extraction. In this way, the network can accurately focus on all the relevant elements of the input, weaken the random noise caused by the network misjudgment, and simultaneously better reduce the computational overhead.

　　The model structure of SUNet is shown in Figure 4. The model is divided into 5 parts: input, encoder, decoder, classification and output. The model's input is the size of the remote sensing image sample c × h × w, where h and w are the height and width of the sample data, set to 512 × 512, and c is the number of input channels. In this paper, multi-band data were used to extract winter wheat. In addition to the four bands of the GF-2 image, two features of NDVI and $NDVI_{increase}$ were added, so c was set to 6. Each layer of the encoder structure contains two 3 × 3 convolutions, each followed by a BN layer and a ReLU layer. A 2 × 2 maximum pool downsampling was used to obtain shallow features. In the decoder part, bilinear interpolation was used for upsampling, and Shuffle Attention optimized the shallow features obtained in the encoder stage through jump links. Then, the image information was restored by splicing and fusion with the deep features extracted by upsampling. The classifier uses a 1 × 1 convolution to output the multi-channel features, and the output result contains n channels. Since this paper dealt with binary classification, specifically the categorization of remote sensing images into winter wheat and non-winter wheat classes, the value of n was set to 2. Finally, the softmax function calculated the mapping of each pixel value of the classification feature graph to the real number [0prime1]. The output image is a single-channel image of the same size as the input image. Each pixel value represents the category to which it belongs, 0 for non-winter wheat and 1 for winter wheat.

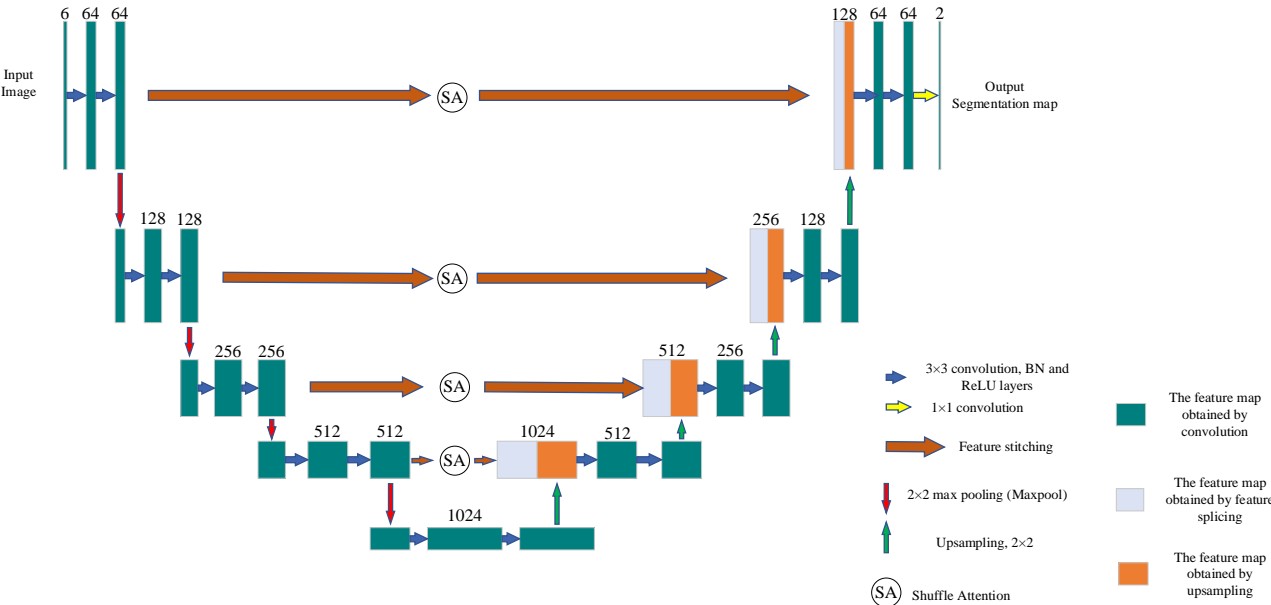

**Figure 4.** SUNet network structure diagram.

### 2.3.1. Batch Normalization

Batch normalization is a neural network optimization method proposed by Ioffe et al. [36]. Normalization can make scattered data regular, so the neural network can better learn the rules between the data. BN processing calculates the variance and mean of each batch of data. In the forwarding transmission of each batch of data, the input data of each layer is normalized by the recorded mean and variance. This ensures that the input distribution at each network layer is the same. It can also effectively improve the convergence speed of the neural network, prevent the gradient from disappearing and alleviate the overfitting phenomenon. This paper improved the UNet model by adding the BN layer after the convolution layer of each network layer. This can improve the network's generalization ability, standardize the network layer's data input, effectively prevent the disappearance of the gradient, accelerate the convergence speed and alleviate overfitting to improve winter wheat classification accuracy. The basic structure of the convolution network added to the BN layer is shown in Figure 5.

### 2.3.2. Shuffle Attention (SA)

The SA mechanism was proposed by Zhang et al. [37] and others from Nanjing University. Firstly, the module uses group convolution to group different input feature maps to reduce computation. Then, channel segmentation is used for each data group. One part is used for spatial attention calculation, and the other participates in channel attention calculation. Concat fuses the feature information of the same group, and finally the sub-feature images obtained through different channel attention mechanisms are reconstructed by channel shuffle operation. Spatial attention uses group normalization (GroupNorm, GN) [38] to obtain information on the spatial dimension. Channel attention uses this global average pooling [39] operation to generate channel-related statistics. Then, the parameters are used to scale and translate channel vectors and generate channel and spatial feature representations. Without increasing the model's size, the UNet network is improved in this paper. In the jump phase, the Shuffle Attention module is added to optimize the features of the coding phase. Then, it is spliced with the features obtained by upsampling to make the neural network pay more attention to the useful features and restrain the useless parts in the decoding process. Thus, the convolution network improves the recognition accuracy of winter wheat in high-resolution images.

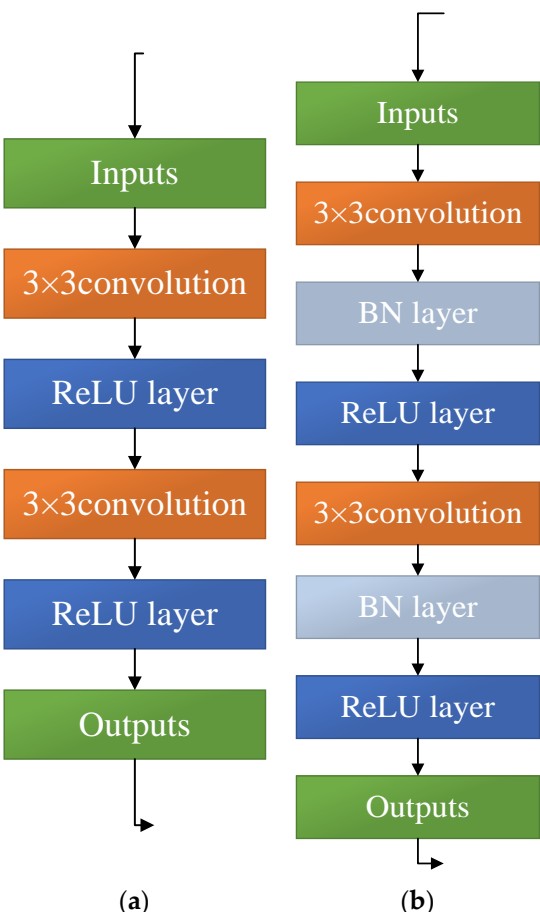

**Figure 5.** Structure comparison. (**a**) UNet original convolutional layer structure; (**b**) UNet convolutional layer structure with added BN layer.

### 2.4. Experimental Setup

#### 2.4.1. Lab Environment

The programming language is Python. In addition, the NumPy library, GDAL library, PIL library and Matplotlib library are used to realize data processing and visualization. This paper implements data enhancement processing, construction, training, testing, and results in analyzing the SUNet neural network based on Baidu's open-source deep learning framework, PaddlePaddle, on the AI studio platform. The detailed experimental configuration environment is shown in Table 2.

**Table 2.** Experimental environment (AI studio) configuration information.

| Surroundings | Name |
|---|---|
| Deep Learning Framework | PaddlePaddle-gpu 2.2.0 |
| Programming language | Python 3.7 |
| CPU | Intel$^{®}$ X$^{®}$(R) Gold 6271C (2.6 Hz) |
| GPU | Tesla V100 (32 G) |
| RAM | 32 G |
| Hard disk | 100 G |

#### 2.4.2. Data Pre-Processing

Due to the influences of factors such as satellite altitude, atmospheric conditions, and observation environment, remote sensing images often exhibit distortions and information errors [40]. Therefore, it is necessary to preprocess the remote sensing images before further analysis. In order to obtain more authentic and reliable remote sensing images, a series

of preprocessing operations were conducted on the selected high-resolution GF-2 satellite imagery using ENVI 5.3 software. These operations included radiometric calibration, atmospheric correction [41], orthorectification, and image fusion [42]. Through these preprocessing steps, distortions caused by sensor and environmental factors were mitigated, leading to improved accuracy and reliability of the images.

Additionally, to enhance the extraction effectiveness of winter wheat information and improve data visualization, the band combination and stretching techniques were applied to the remote sensing images. The band combination process fused data from different spectral bands, providing more comprehensive information, while preserving the 8-bit data to ensure data accuracy. The stretching process adjusted the brightness and contrast of the images using histogram equalization, resulting in enhanced visibility of details. These processing techniques enabled us to achieve better visual effects and laid a solid foundation for subsequent winter wheat patch extraction and analysis.

Deep learning requires a large number of data samples. Due to the diversity of remote sensing images and the continuous changes in the growth stages of winter wheat, there is a limited availability of publicly available datasets specifically related to winter wheat. Therefore, the high-resolution GF-2 images in this study were classified and annotated to create a dataset suitable for deep learning training. The images were labeled into two categories, winter wheat and non-winter wheat. The category features are shown in Table 3. The non-winter wheat in Table 3 refers to water bodies, buildings, roads, and other vegetation categories; the color changes in water bodies, buildings, roads, etc., in the front and rear images are small, and the changes in other vegetation are opposite to winter wheat and gradually turn green. From the pictures in Table 3, it can be seen that the color of winter wheat is green in the image of March, and brown in the image of May, with obvious changes and clear texture features. In order to obtain a more accurate winter wheat dataset, this study manually annotated the winter wheat on overlaid images from four different sources: (1) Training images, which were GF-2 satellite imagery from 5 March 2020; (2) Winter wheat mask data extracted using the random forest research method [34]; (3) GF-2 satellite imagery from the same area during 25 May 2020, when winter wheat underwent noticeable color changes compared with the training images; (4) Survey data from Lankao County. Based on the survey data, this paper's GF-2 satellite imagery from March was annotated by combining the winter wheat mask regions and the variations of winter wheat in different growth stages in the imagery.

**Table 3.** Features of remote sensing image categories.

| Tag Category | Tag Value | March Features | May Features |
|---|---|---|---|
| Non-winter wheat | 0 |  |  |
| Winter wheat | 1 |  |  |

### 2.4.3. Loss Function

Because the planting range of winter wheat is relatively concentrated, and some areas are scattered, the distribution of categories are inevitably not balanced in the sample dataset.

Some samples have a large proportion of winter wheat, while others have more non-winter wheat. The traditional two-category cross-entropy loss function calculation formula is shown in Equation (1),

$$\text{CE}(r,p) = -r\log p - (1-r)\log(1-p) = \begin{cases} -\log p, r = 1 \\ -\log(1-r), r = 0 \end{cases} \tag{1}$$

where r represents the true sample value and p represents the predicted class prediction probability during model training. According to the Equation (1), for positive samples, the higher the probability, the smaller the loss value; for negative samples, the lower the probability, the lower the loss value. This aligns with the principle that the more accurate the prediction, the lower the loss. However, when the sample category distribution is uneven, the loss value produced by many class samples covers up the loss value produced by a small number of class samples. This causes the gradient decline direction of the loss function to be inconsistent with the direction this paper expects to achieve. Based on this problem, Lin et al. [43] proposed an improved focus loss function based on the cross-entropy loss function (Focal Loss). The formula is shown in Equation (2).

$$\text{FL} = \begin{cases} -\alpha(1-p)^{\gamma}\log p, y = 1 \\ -(1-\alpha)p^{\gamma}\log(1-p), y = 0 \end{cases} \tag{2}$$

where $\alpha$ and $\gamma$ are adjustable variables. It can be seen from the formula that the weight of the loss value of the classification result is increased compared with the two-classification cross-entropy loss function. As a result, the loss weight of the positive sample is low, and the weight loss of the negative sample is high, which can effectively solve the problem of sample imbalance. Therefore, this paper chose Focal Loss as the loss function to calculate the loss value of the classification results.

### 2.4.4. Training Process

The training parameters used in this paper are shown in Table 4. After many experiments, the epoch value was set to 200, the batch size was set to 4, the learning rate was set to 0.0001, and the cosine annealing decay attenuated the learning rate. In order to update the parameters more accurately, it is necessary to optimize the network parameters. The Adam optimization algorithm performs excellently and is widely used in neural network optimization. Therefore, this paper used the Adam optimization algorithm to optimize the model.

**Table 4.** Details of training parameters.

| Hyperparameter Name | Parameter Value |
| --- | --- |
| epoch | 200 |
| Batch size | 4 |
| Initial learning rate | 0.0001 |
| Learning rate decay method | Cosine annealing decay |
| Optimizer | Adam |

This paper used the parameters in Table 4 to train and test the built network model. The training process is as follows, and the process is shown in Figure 6:

(1) The super parameter initialization of the network model training process is determined.
(2) The training data are input into the network model for forwarding calculation. The features are extracted by the convolution layer, pooling layer and deconvolution layer hidden in the network coding and decoding part. Finally, all the sample pixels are classified in the classification layer to obtain a set of predictive values xp.
(3) The focal loss function is used to calculate the error between the predicted value xp and the true value. If the error meets the target requirement, the training is completed. Otherwise, the training is continued.

(4)  The loss value is derived by the Adam optimizer, and the parameters are back-propagated to realize the parameter update of the SUNet network model, thus reducing the loss value.

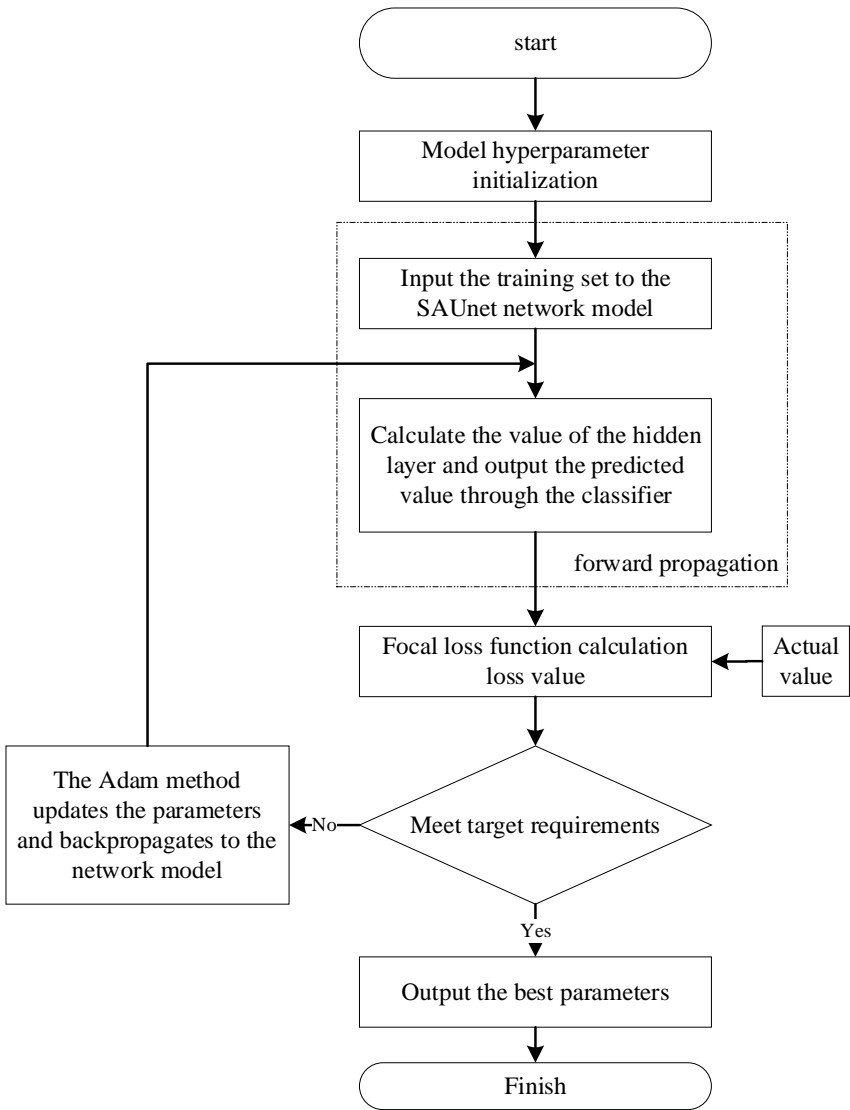

**Figure 6.** Neural network training process.

2.4.5. Evaluation Metrics

This paper quantitatively evaluated the classification accuracy of the network model used in this paper by employing metrics commonly utilized in semantic segmentation. These metrics include mean intersection ratio (mIou), overall classification accuracy (OA), precision (precision), recall (recall), F1 score, and kappa coefficient.

The calculation formula of the average intersection and union ratio is shown in Equation (3). It represents the ratio of the intersection and union of the actual and predicted values.

$$\text{mIou} = \frac{1}{k+1} \sum_{i=0}^{k-1} \frac{P_{ij}}{\sum_{j=0}^{k-1} P_{ij} + \sum_{j=0}^{k-1} P_{ji} - P_{ii}}. \tag{3}$$

The formula for calculating the overall classification accuracy is shown in Equation (4). It represents the ratio of correctly classified pixels to the total number of pixels.

$$\text{OA} = \frac{\sum_{i=0}^{k-1} P_{ii}}{\sum_{i=0}^{k-1} \sum_{j=0}^{k-1} P_{ij}}. \tag{4}$$

The formula for calculating the accuracy rate is Equation (5). It represents the ratio of the correctly classified pixels of winter wheat to the total number of predicted winter wheat pixels.

$$\text{precision} = \sum_{i=0}^{k-1} \frac{P_{ii}}{\sum_{j=0}^{k-1} P_{ij}}. \tag{5}$$

The formula for the recall rate is shown in Equation (6). It represents the ratio of the number of correctly classified pixels of winter wheat to the total number of pixels of real winter wheat.

$$\text{recall} = \sum_{j=0}^{k-1} \frac{P_{jj}}{\sum_{i=0}^{k-1} P_{ij}}. \tag{6}$$

The F1 score calculation formula is shown in Equation (7). It is a comprehensive evaluation index that can consider the accuracy of the classification model and the recall rate.

$$\text{F1} = 2 \times \frac{\text{precision} \times \text{recall}}{\text{precision} + \text{recall}}. \tag{7}$$

The kappa coefficient is used to measure the classification effect of winter wheat, and its calculation formula is shown in Equation (8),

$$\text{kappa} = \frac{\text{OA} - \frac{\sum_{x=0}^{k-1} \left[ \sum_{i=1}^{k-1} P(i,x) * \sum_{j=1}^{k-1} P(x,j) \right]}{\left[ \sum_{i=0}^{k-1} \sum_{j=0}^{k-1} P(i,j) \right]^2}}{1 - \frac{\sum_{x=0}^{k-1} \left[ \sum_{i=1}^{k-1} P(i,x) * \sum_{j=1}^{k-1} P(x,j) \right]}{\left[ \sum_{i=0}^{k-1} \sum_{j=0}^{k-1} P(i,j) \right]^2}}, \tag{8}$$

where $P$ represents the calculated confusion matrix. When $i \neq j$, $P_{ij}$ represents the number of pixels in which class $I$ is mistakenly divided into class $j$, $P_{ij}$ represents the number of pixels in class $j$ misclassified into class $I$, $P_{ij}$ represents the number of pixels correctly classified, and k represents the number of categories.

### 2.4.6. Experimental Description

This study conducted multiple sets of experiments for comparative analysis to verify the effectiveness and accuracy of winter wheat extraction in high-resolution remote sensing images. These experiments included popular semantic segmentation models, such as UNet, SegNet, U2-Net, and DeeplabV3+. Two network architectures were designed for ablation experiments to compare the impact of batch normalization (BN) layer and Shuffle Attention (SA) mechanism on performance improvement. The first one is Net-1, which includes the BN layer but does not have the SA mechanism. The second one is Net-2, which does not include the BN layer but has the SA mechanism.

## 3. Experimental Section and Results
### 3.1. Datasets

The authors produced the dataset used in this paper. This dataset has been published in Kaggle and linked in the data availability statement. This winter wheat dataset is a multi-band remote sensing image classification dataset. After analyzing the NDVI changes during the growth period of winter wheat, it was observed that the NDVI increase in winter wheat underwent significant variations compared with other land types. Moreover, both the NDVI values and the magnitude of NDVI$_{\text{increase}}$ during the growth stage directly

impacted the accuracy of winter wheat extraction. Therefore, first, this paper inverted the NDVI on the GF-2 image using the ENVI software. Second, the NDVI$_{increase}$ extracted by the method proposed by Wang et al. [44] was resampled according to the projection method, and the spatial resolution of GF-2 and data were normalized. Finally, two bands were added to the original band of the GF-2 image, namely NDVI and NDVI$_{increase}$ extracted from the Landsat 8 image. The NDVI$_{increase}$ formula [34] is shown in Equation (9). NDVI$_{min}$ is the minimum NDVI composite image of the image set from September to November in the early sowing period. NDVI$_{max}$ is the maximum NDVI composite image of the image set from December to March of the following year.

$$NDVI_{increase} = \frac{NDVI_{max} - NDVI_{min}}{|NDVI_{min}|}. \tag{9}$$

After the remote sensing image processing in Section 2.4.3 and the NDVI inversion and data normalization processing in the previous paragraph, this paper used the GDAL library in the Python language to read the raster image data and segmented the classified image and the labeled image in the same way. When the segmentation was completed, the labeled image was converted into a single-channel PNG format image. After the above process, a total of 648 groups of data samples was collected, and each image size was 512 × 512. These images were allocated in an 8:2 manner, in which 80% of the data was used for image training and 20% for classification accuracy verification. The winter wheat sample data is shown in Figure 7.

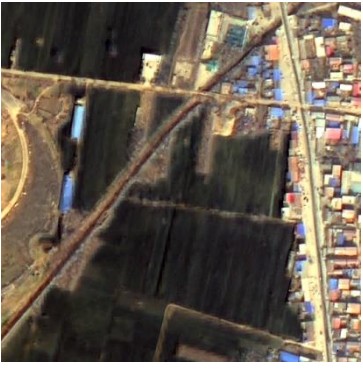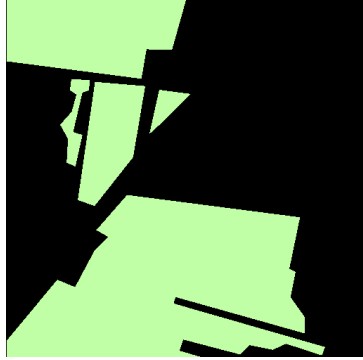

**Figure 7.** Sample of winter wheat (Green area is winter wheat, black area is non-winter wheat).

Data Augmentation

In deep learning, the network model is prone to overfitting when the sample data are limited. This study used a small and limited winter wheat sample dataset. To enhance the generalization ability of the network model during the training process and avoid overfitting, this paper used data augmentation techniques to expand the sample dataset without altering the labeling results.

Specifically, various data augmentation strategies were applied, including horizontal flipping, vertical flipping, 90° and 180° rotation, as well as diagonal flipping (diagonal mirroring), as shown in Figure 8. These operations generated a larger set of variant sample data while preserving the original sample's labeling results. These augmentation techniques aimed to increase the samples' diversity, enabling the network model to learn the winter wheat features under different variations, thus improving the winter wheat patch boundary recognition and extraction accuracy.

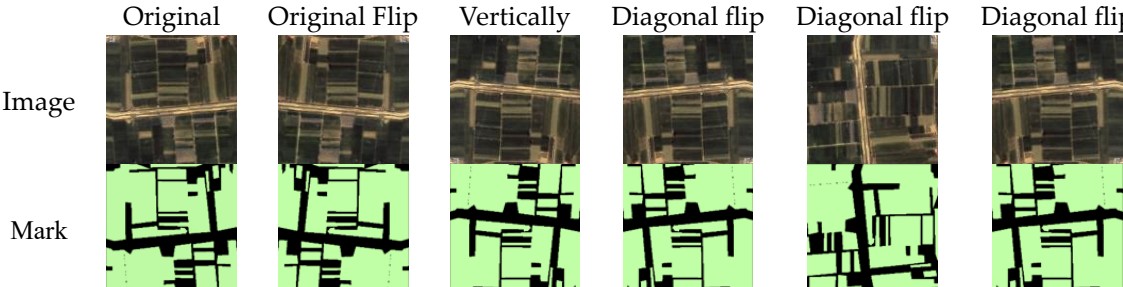

**Figure 8.** Data augmentation example.

*3.2. Results*

3.2.1. Comparative Test Result

The verified results are shown in Figure 9, and the evaluation indicators are shown in Table 5. Table 5 shows that after adding the BN layer and the Shuffle Attention mechanism to the UNet network structure, the indicators of the structure are significantly improved compared with the original network—MIou increased by 2.53%, OA increased by 1.18%, and the classification accuracy of winter wheat was improved by 1.59%. The classified recall rate of winter wheat increased by 2.1%, the F1 score by 1.85%, and the kappa coefficient by 2.72%. Table 5 also shows that compared with the classical image semantic segmentation networks SegNet, U$^2$-Net and Deeplabv3+, SUNet performs better in all indicators. Regarding the comparison of the size of the model, after adding the BN layer and the SA mechanism, the size of the model after SUNet training is only 0.1 m bigger than that of the UNet network. In addition, it is much smaller than the model size of SegNet, U$^2$-Net and Deeplabv3+.

**Table 5.** Evaluation index results of each network model.

| Model<br>Index | Unet | SegNet | U$^2$-Net | Deeplabv3+ | SUNet |
|---|---|---|---|---|---|
| mIou | 0.9261 | 0.9348 | 0.9437 | 0.9422 | 0.9514 |
| OA | 0.9663 | 0.9703 | 0.9746 | 0.9739 | 0.9781 |
| precision | 0.9460 | 0.9490 | 0.9596 | 0.9560 | 0.9619 |
| recall | 0.9497 | 0.9597 | 0.9619 | 0.9633 | 0.9707 |
| F1 | 0.9478 | 0.9543 | 0.9608 | 0.9596 | 0.9663 |
| kappa | 0.9229 | 0.9324 | 0.9419 | 0.9403 | 0.9501 |
| model size | 51.1 M | 113 M | 168 M | 175 M | 51.2 M |

Figure 9 shows the test results of each network model on a partial verification set. As can be seen from the figure, the segmentation effect of the UNet and SegNet network models is poor. They have more misdivided pixels, and the effect of boundary segmentation is poor. The segmentation effect of UNet is slightly worse than that of SegNet. The network depth of the SUNet model is much smaller than that of the U2-Net and Deeplabv3+ networks. However, it can be seen from Figure 9 that its segmentation effect is better than that of the U2-Net and Deeplabv3+ networks, and the boundary segmentation effect is also better. The experimental results show that the segmentation effects of the five kinds of networks are as follows: UNet < SegNet < Deeplabv3+ < U$^2$-Net < SUNet.

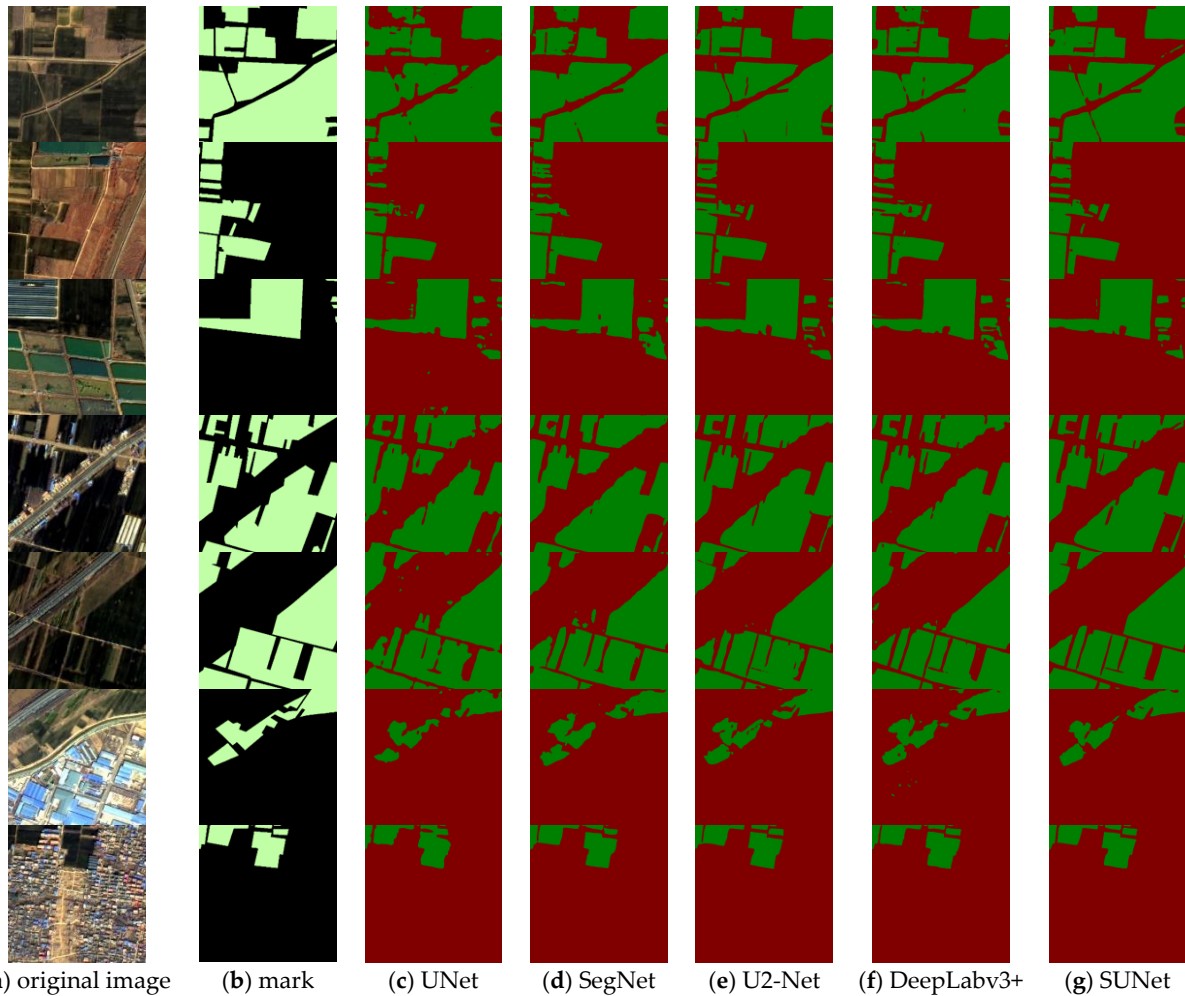

| (**a**) original image | (**b**) mark | (**c**) UNet | (**d**) SegNet | (**e**) U2-Net | (**f**) DeepLabv3+ | (**g**) SUNet |

**Figure 9.** Experimental comparison results of winter wheat image dataset (Green areas are winter wheat, black areas and red areas are non-winter wheat).

### 3.2.2. Ablation Experiment Results

The extraction results after ablation experiments are shown in Figure 10, and the accuracy evaluation results are shown in Table 6. As can be seen from Table 6, whether it is Net-1 after adding the BN layer alone or Net-2 after adding the SA mechanism alone, all indicators have been improved compared with the original UNet network. However, the accuracy evaluation indexes of Net-1 and Net-2 are not as high as those of SUNet. Figure 10 shows the extraction results of the Net-1 and Net-2 network models on some verification sets. It can be seen from the graph that the segmentation effect of Net-1 and Net-2 are better than that of UNet, but both are slightly worse than that of SUNet.

**Table 6.** Evaluation index results of ablation experiment.

| Model / Index | UNet | Net-1 | Net-2 | SUNet |
|---|---|---|---|---|
| mIou | 0.9261 | 0.9454 | 0.9361 | 0.9514 |
| OA | 0.9663 | 0.9753 | 0.9710 | 0.9781 |
| precision | 0.9460 | 0.9604 | 0.9516 | 0.9619 |
| recall | 0.9497 | 0.9634 | 0.9588 | 0.9707 |
| F1 | 0.9478 | 0.9619 | 0.9552 | 0.9663 |
| kappa | 0.9229 | 0.9436 | 0.9337 | 0.9501 |

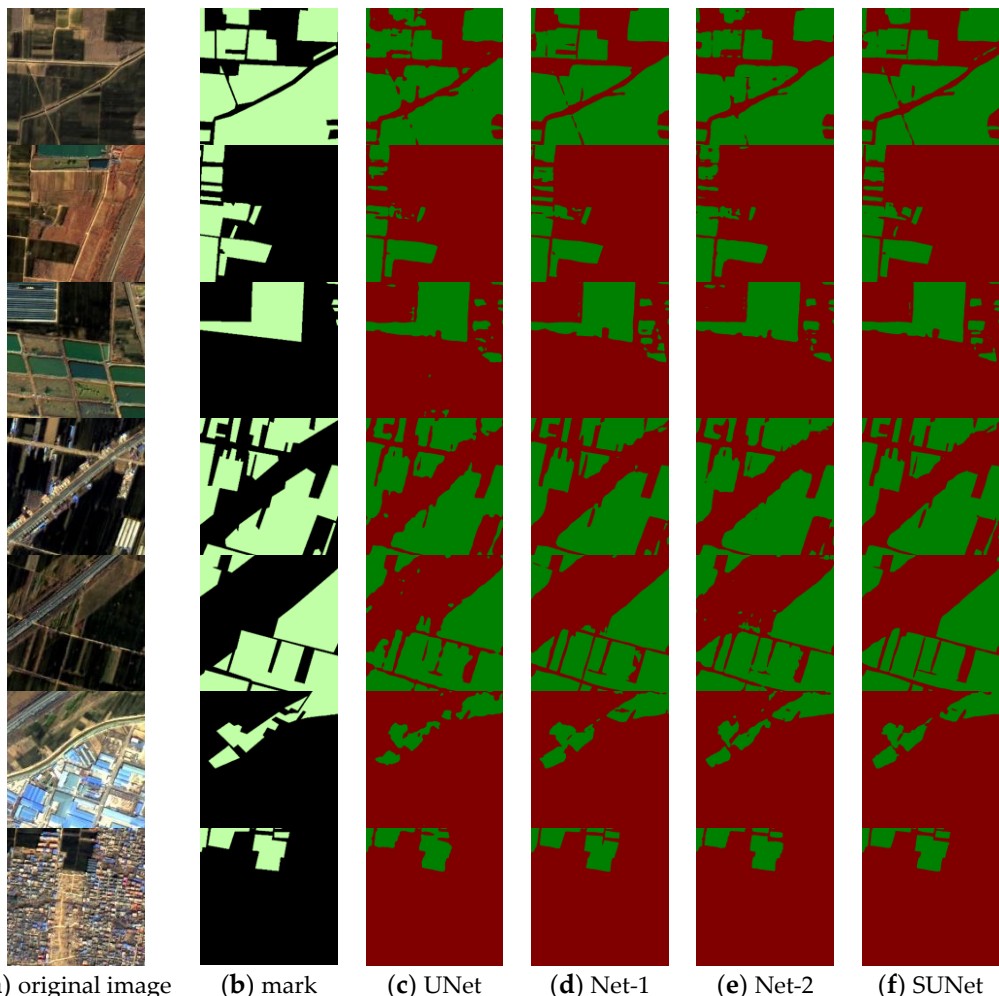

(**a**) original image    (**b**) mark    (**c**) UNet    (**d**) Net-1    (**e**) Net-2    (**f**) SUNet

**Figure 10.** Extraction results of ablation experiment (Green areas are winter wheat, black areas and red areas are non-winter wheat).

## 4. Discussion

### 4.1. Comparative Test Discussion

The method used in this paper can identify and extract winter wheat in the western region of Lankao County, Kaifeng City. In addition, it can be seen from Table 5 and Figure 9 that SUNet's indicators have been improved. Compared with the UNet network, the SUNet network has a better recognition effect and higher recognition accuracy. Compared with the U$^2$-Net network proposed by Qin et al. [45] and the Deeplabv3+ network proposed by Chen et al. [46], the depth of SUNet is smaller, and the boundary segmentation effect of winter wheat is better. This shows that adding the BN layer and the SA mechanism can effectively alleviate the over-fitting phenomenon and shield some useless information to improve winter wheat recognition accuracy. From Figure 9, it can also be concluded that deep learning has a certain fault tolerance. Since the label is artificial, error and omission may inevitably occur, and the selection of samples is not completely correct. In the experiment, the label data were learned through deep learning, and the missing parts were sometimes identified. For example, a small part of winter wheat is not labeled on the third dataset's right side. However, all network models can be identified effectively. The fifth data group can be mistakenly classified as winter wheat, but the SUNet model performs well. This shows that the deep learning method has a strong fault tolerance, which can effectively reduce errors and improve target classification accuracy.

*4.2. Ablation Experiment Discussion*

From the results of Figure 10 and Table 6, the ablation experiment shows the following three points: Net-1 and Net-2 have a better extraction effect than UNet; Net-1's extraction effect is better than that of Net-2. Finally, SUNet's extraction effect is better than that of Net-1 and Net-2. Firstly, by analyzing the comparative results between Net-1 and UNet, it can be observed that adding a BN layer can effectively prevent overfitting and improve segmentation accuracy. This is because including a BN layer allows the model to depend on statistical features during the training process, such as the mean and variance of each batch, rather than solely relying on input features. This approach can effectively control the model's complexity, thus avoiding overfitting. The comparison results between Net-2 and UNet show that the classification accuracy is improved after adding an SA mechanism. This is because the SA mechanism can enable the network to filter out irrelevant feature information, enhance the model's focus on regions of interest, and pay more attention to useful features, thereby improving classification accuracy. In addition, the comparison results of Net-1 and Net-2 indicate that adding a BN layer alone in UNet is more effective than adding an attention mechanism alone. This may be because the effect of the BN layer is more direct. In comparison, the advantage of the attention mechanism may not be as good as that of the BN layer when computing resources are limited. Finally, by analyzing the comparison results of SUNet, Net-1, and Net-2, it can be concluded that the performance of adding both a BN layer and attention mechanism in UNet is better than that of adding only a BN layer or attention mechanism separately. This is because a BN layer and attention mechanism can complement each other, improve the model's performance at different levels, and compensate for each other's shortcomings. This is also why SUNet has better feature extraction and accuracy evaluation indicators than other networks in this paper.

## 5. Conclusions

This work studied the recognition of winter wheat based on a convolutional neural network in the cloud environment. This paper improved the UNet model, proposed the SUNet model, and used the AI Studio cloud platform to study the application of convolution neural networks in winter wheat recognition on GF-2 high resolution. The conclusions are as follows: the high-performance environment equipped with an AI Studio cloud platform can provide efficient computing support for deep learning methods to identify winter wheat on high-resolution images. SUNet adds a BN layer and Shuffle Attention to UNet and uses the focus loss function to train the winter wheat dataset of multi-band GF-2 images with NDVI and $NDVI_{increase}$. This can effectively optimize the features. Thus, the precision of the winter wheat recognition on high-resolution images can be improved. The mIou calculated based on the confusion matrix is 0.9514. In addition, the overall classification accuracy, the F1 score of complete accuracy and the kappa coefficient are 0.9781, 0.9663 and 0.9501, respectively. Deep learning has a certain degree of fault tolerance, and unlabeled and mislabeled areas can be identified by training a convolution network. The work presented in this paper still has certain limitations. The dataset used in this study is suitable for extracting spatial information of winter wheat during a specific period. Currently, SUnet exhibits superior performance only in the task of spatial information extraction of winter wheat during specific periods. However, its performance in winter wheat change detection tasks, which involve detecting the variations of winter wheat at different time points, remains unclear. Therefore, further research and evaluation are needed to determine the performance and applicability of SUnet in the task of winter wheat change detection. The next step is to combine the time-series variations of the winter wheat growth cycle and extract the spatial distribution information of winter wheat during different periods.

**Author Contributions:** K.Z., R.M. and Y.Y. conceived the idea and proposed the method. L.L. provided suggestions for the research and helped edit the thesis. Z.Z., T.R. and M.Y. improved the quality of the manuscript and completed revisions. All authors have read and agreed to the published version of the manuscript.

**Funding:** The research was funded by The Henan Science and Technology Project (202102210381), the Major Science and Technology Project of Kaifeng City (18ZD007), the Key Technology Project of Kaifeng City (2001001), and the Science and Technology Project of Henan Province (222102210061).

**Data Availability Statement:** The dataset used to support the findings of this study is provided at https://www.kaggle.com/datasets/zhangzzzy/winterwheatdataset accessed on 25 July 2022.

**Conflicts of Interest:** The authors declare no conflict of interest.

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
