# Peer review of "Research on SUnet Winter Wheat Identification Method Based on GF-2"

_remotesensing, doi:10.3390/rs15123094_

Round 1
Reviewer 1 Report (Previous Reviewer 1)
Point 1: P119: It would be advisable to include the location of Henan Province within China on the map of the study area. This would be particularly helpful for international readers or those who are not familiar with the geography of China.
Point 2: P247-248: If there are references to relevant literature for other operations, please include them in the citation as well.
Point 3: P263:"We're" instead of "are" should be used. Please check the tense throughout the entire text. Present tense is generally used to describe general facts. Specific observations and the work done in this paper are usually described using past tense.
Point 4: P268:The random forest method used needs to cite relevant literature
Point 5: P276:After creating the dataset, data augmentation techniques were applied. So, would it be more appropriate to include the data augmentation section within the dataset chapter?
Point 6: P472: [Error! Reference source not found.]
Minor editing of English language required.
Author Response
Please see the attachment.
We gratefully thank editor and all reviewers for their time spend making their constructive remarks and useful suggestions, which has significantly raised the quality of the manuscript and has enables us to improve the manuscript. Attached are the point-by-point responses to the reviewers' comments.

Reviewer 2 Report (New Reviewer)
Dear All,
Please, see the posted comments regarding scientific and linguistic issues.
Sincerely Yours,

Author Response
Please see the attachment.
We gratefully thank editor and all reviewers for their time spend making their constructive remarks and useful suggestions, which has significantly raised the quality of the manuscript and has enables us to improve the manuscript. .We have included a point-by-point response to the reviewers' comments in a PDF file.

Reviewer 3 Report (New Reviewer)
The experimental work presented in the Manuscript, entitled „ Research on SUnet winter wheat identification method based on GF-2 " is interesting research with some promising results. The article reports SUNet network can effectively improve winter wheat recognition accuracy in multi-band GF-2 images. Furthermore, with the support of a cloud platform, it can provide data guarantee and computing
support for winter wheat information extraction, there are modifications that should be included in order to enhance the final manuscript for the readers.
Abstract
1. Abstract is more description and it does not support by enough digital results.
2. Abstract should include the introduction sentient to present the important of this study.
3. Please remove the sentence from lines 20 to 21 (The SUNet network used in this study is superior to the original UNet, Segnet, deeplabv3 + and other results in various indicators).
Introduction
4. Line 91. This sentence (In summary, the main contributions of this paper are as follows) need to modified to the objectives of this study…………..
5. Please remove the sentences from 99 to 103. This is structure of the manuscript.
Methodology
6. Please increase the size and resolution of figure 1.
7. Line 137 and line 146 Change in this paper to in this study.
Results and discussion were good written
Please, write the practical applications of your work in a separate section, before the conclusions and provide your good perspectives.
Conclusion
Please write about the limitations of this work in details in conclusion section.
Minor editing of English language required
Author Response
Please see the attachment.
We gratefully thank editor and all reviewers for their time spend making their constructive remarks and useful suggestions, which has significantly raised the quality of the manuscript and has enables us to improve the manuscript. Attached are the point-by-point responses to the reviewers' comments.

This manuscript is a resubmission of an earlier submission. The following is a list of the peer review reports and author responses from that submission.
Round 1
Reviewer 1 Report
The use of high-resolution remote sensing data to identify crop types has natural advantages over traditional mid-to-high-resolution data e.g. sentinel. The topic selection of this manuscript has certain significance. However, there are many deficiencies in the data description, method design, and discussion parts of this manuscript. It is recommended to reconsider after major revision.
major problems:
According to the description in the method chapter, it seems the innovation of this article lies in the introduction of the modules including batch normalization layer and Shuffle Attention. For an article based on deep learning methods, an ablation experiment may be needed to illustrate what role the module plays.
The data description in 2.5.2 Production of datasets is confusing. The study performed a layer-stacking on three-band data. But it is also necessary to explain how many bits of data are saved? 8-bit or 16-bit? Whether performed rendering, and what is the method of rendering or stretching.
The Discussion section is too simplistic to provide enlightening information.
The expression of the article requires systematic revision and polishing.
Minor problems:
L38: I don't think it's accurate to say that CNNs are widely used in NLP.
L59: “However, the current research mostly uses sentinel data and landsat data, and seldom uses Chinese high-score data.” Please explain why? In addition, what is Chinese high-score data? Do you mean Gaofen data?
L101: Please add a space after coma.
L270: what is “real remote sensing images”?
L273: “Deep learning requires sample data as a driver.” The expression is not formal enough.
L283: “the image color performance of winter wheat changed significantly with the image to be classified”. This sentence is difficult to understand.
L290: “GF-2 images are 4, 3, and 2 bands” which product of GF-2, PMS or WFV, what are the name of the bands, red, green, blue, and NIR?
typos in in L97.
Reviewer 2 Report
Major comments:
1) Authors use definition "improved CNN", "improved UNet" as alternative to SUNet, that is confusing and complex to identify where are general sentences and text exactly about SUNet. Please, one-time predefine SUNet and use this definition for description of your own solution.
2) The description of dataset must be strongly improved. The input of SUNet has 6 channels, but description mentions 2xNDVI. Meanwhile, NDVIincrease - how long difference in days not mentioned. As well as, Kaggle repository does not contain dataset metadata, - all data must be filled for long usage of dataset. Additionally, Kaggle dataset must be mentioned in the text not only in the end of article. Kaggle has 2xPNG (RGB images, labels) and .npy (content unclear). PNG is not spectral data. RGB or spectral data was applied? How large is dataset collection?
Minor comments:
[Lines 45, 56] What are good results? Please, reformulate considering scientific practice - only facts.
[Lines 19-20 and 78-79] Different loss functions are mentioned as applied.
[Lines 100-101] How could it be launched in 19 Aug., if the first launch had been in 21 Aug.? How it can be launched two times? Please correct sentence.
[Table 1] Symbol '~' means changable.
[Table 1] Considering to sources: https://www.eoportal.org/satellite-missions/gaofen-2#mission-capabilities, https://www.satimagingcorp.com/satellite-sensors/gaofen-2/
there are PAN = 0.81m and MS = 3.24 m
[Lines 238-239] Similar sentences with contradiction (dif. tables 2 & 4)
[Lines 306-308] "accuracy verification" - validation or testing?
[Lines 75-76, 183-186, 204-205, 402-404] Considering to experiment, it can be concluded only that SUNet is better than UNet. Experiment does individually evaluate BN impact on all possible UNet models, similar situation with SA. Another experiment design must be planned to prove BN and SA impact on UNet. As well as, BN is common practice to improve CNN training, that is proved by original authors using math.
[Lines 405-407] Figure 6 depicts augmentation.
[Lines 409-410] It is subjective conclusion based on image. Must be based on statistics.
P.S. Comments with questions: the explanation must be provided in article, it is not individual question.
Reviewer 3 Report
Hi Authors.
I read through the paper and it must be restructured properly. The paper must pass through the hands of an english editor. The paper must have the following
Abstract, Introduction, Study Areas, Methodology, Results and Discussion. - The researchers are mixing things (there are equations in the results section and these are supposed to be part of methodology)
- they are using present tense not past tense
- they are mixing between third person and first person reporting. They must stick to one
- some of the sentences are too long
- some of the information that is coming from literature is not referenced.
- the paper is quite novel but it need to properly restructured.
I hope this will help improve their paper.
Regards
Round 2
Reviewer 1 Report
The English expressions of many statements are not sufficiently native.
P31: It is better to change "High-precision" to "Accurately".
P39: "As a result of its success in the field of computer vision, it has made significant progress in remote sensing [24, 25] image automatic interpretation." Please make the meaning of this sentence clearer.
P62: "However, the current research mainly uses sentinel and Landsat data and seldom uses Chinese GF data, which is difficult to obtain." You need to explain why it is difficult to access. High cost and no access? This should be explained clearly for people who are not Chinese.
P64: "The detailed information of winter wheat patches in the GF-2 remote sensing image exceeds that of sentinel and Landsat data."
Many of the descriptions like this in the manuscript are quite general and lacking in detail to support them. This sentence should be rephrased to: "The GF-2 PMS data is capable of providing optical images at 0.8m resolution, which greatly exceeds that of Sentinel (10m) and Landsat (30m), thus providing a wealth of information on winter wheat e.g. texture, field shape."
Reviewer 3 Report
Hi.
I have attached the paper with my comments. I think the paper need proper restructuring and must pass through the hands of an english editor.
